# Trial Assay for Safe First-Aid Protocol for the Stinging Sea Anemone *Anemonia viridis* (Cnidaria: Anthozoa) and a Severe Toxic Reaction

**DOI:** 10.3390/toxins14010027

**Published:** 2022-01-01

**Authors:** Ainara Ballesteros, Janire Salazar, Macarena Marambio, José Tena, José Rafael García-March, Diana López, Clara Tellez, Carles Trullas, Eric Jourdan, Corinne Granger, Josep-Maria Gili

**Affiliations:** 1ICM-CSIC-Institute of Marine Sciences, Department of Marine Biology and Oceanography, Passeig Marítim de la Barceloneta 37-49, 08003 Barcelona, Spain; jsalazar@icm.csic.es (J.S.); marambio@icm.csic.es (M.M.); gili@icm.csic.es (J.-M.G.); 2IMEDMAR-UCV-Institute of Environment and Marine Science Research, Universidad Católica de Valencia SVM, C. Explanada del Puerto S/n, Calp, 03710 Alicante, Spain; josetena@ucv.es (J.T.); jr.garcia@ucv.es (J.R.G.-M.); diana.lopez@ucv.es (D.L.); clara.tellez@ucv.es (C.T.); 3ISDIN, Innovation and Development, C. Provençals 33, 08019 Barcelona, Spain; carles.trullas@isdin.com (C.T.); eric.jourdan@isdin.com (E.J.); corinne.granger@isdin.com (C.G.)

**Keywords:** cnidarian venom, cnidocyst discharge, cnidocyte, marine knowledge, risk prevention, seawater, sting, vinegar

## Abstract

*Anemonia viridis* is an abundant and widely distributed temperate sea anemone that can form dense congregations of individuals. Despite the potential severity of its sting, few detailed cases have been reported. We report a case of a severe toxic reaction following an *A. viridis* sting in a 35-year-old oceanographer. She developed severe pain, itching, redness, and burning sensation, which worsened one week after treatment with anti-inflammatories, antihistamines and corticosteroids. Prompted by this event, and due to the insufficient risk prevention, lack of training for marine-environment users, and lack of research into sting-specific first-aid protocols, we evaluated the cnidocyst response to five different compounds commonly recommended as rinse solutions in first-aid protocols (seawater, vinegar, ammonia, baking soda, and freshwater) by means of the Tentacle Solution Assay. Vinegar and ammonia triggered an immediate and massive cnidocyst discharge after their application and were classified as activator solutions. Baking soda and freshwater were also classified as activator solutions, although with a lower intensity of discharge. Only seawater was classified as a neutral solution and therefore recommended as a rinse solution after *A. viridis* sting, at least until an inhibitory solution is discovered.

## 1. Introduction

Cnidarians are recognized as venomous marine animals. The phylum includes five main classes (Hydrozoa, Scyphozoa, Cubozoa, Anthozoa, and Staurozoa), which share a distinctive feature: the presence of cnidocytes [1,2]. The cnidocyte stores a mixture of bioactive toxins in a subcellular-enclosed capsule termed a cnidocyst. Cnidocysts are traditionally classified into three categories: nematocyst, spirocyst, and ptychocyst [2,3]. The nematocyst category is the most varied. Up to thirty types, with different capsule morphologies and tubule patterns, are capable of catching and entangling prey and acting as defense against predators [2,4,5]. Differing from nematocysts, the spirocyst and ptychocyst categories are composed of a single type of cnidocyst each [2]. Spirocysts are used to immobilize prey in most anthozoans, and ptychocysts are used by the subclass Ceriantharia for the creation of protective tubes to live in [4,6].

Cutaneous contact between users of marine environments and cnidarians triggers immediate cnidocyst discharge [1,7]. Some types of cnidocysts are able to penetrate human skin, inoculating the venom and causing severe reactions [7,8]. The toxicity of cnidarian stings varies across species. In Australia, *Chironex fleckeri* and *Carukia barnesi* are responsible for serious health problems and lethal envenomation [9,10]. Fortunately, the majority of cnidarians are considered non-life-threatening species [11], and the severity of their stings depends on various factors (e.g., the composition of the venom, the health conditions of the victim, and/or the cnidocyst types) [8,11,12].

In contrast to the high number of fatalities in tropical and subtropical coastal regions [9,10], species that inhabit the Mediterranean Sea commonly cause low to mild skin reactions and annoyances for beach users [13,14], and in isolated cases, systemic symptoms [15,16,17]. Scyphozoans such as *Pelagia noctiluca* and *Rhizostoma pulmo* are common jellyfish in the Mediterranean basin, and their stings are one of the main reasons for seeking assistance from the rescue services [18,19]. However, other cnidarians, such as sea anemones, are also responsible for severe skin reactions in the Mediterranean basin [17,20,21].

The snakelocks sea anemone *Anemonia viridis* is a temperate anthozoan distributed throughout the Mediterranean Sea and in the northern area of the Atlantic Ocean, from the North Sea to the Azores [22]. *A. viridis* clonemates can cover wide areas of the seafloor, playing an important role as a dominant predator in benthic communities [23]. The species’ complexity has been largely studied, but it is important to highlight that due to the presence of different morphotypes and continuous advances in research, there is no consensus on its taxonomy [24]—in this article, it is referred to as *A. viridis* because it is the most recognized and accepted nomenclature [24]. Five different morphotypes of *A. viridis* and four cryptic species have been described to date [25]. The cnidome, which includes the total complement of cnidocysts within a cnidarian specimen [4,5], comprises four types in adult individuals collected in the North Adriatic Sea; three nematocysts (holotrichous isorhiza, p-mastigophore, and a possible microbasic b-mastigophore) and one spirocyst [26]. Envenomation from sea anemones usually involves mild signs and symptoms of erythema, irregular, and painful plaques and papules [17,20,27,28]; rarely, cases of *A. viridis* stings have been reported [17,20].

The diversity of cnidarian species and the lack of studies providing scientific evidence on the effectiveness of first-aid protocols [29] increase the risk of inadequate treatment of users of marine environments. First-aid protocols are focused on two clear actions [30]: (1) rinsing the affected area with a safe rinse solution to remove remaining tissues (e.g., pieces of tentacles) and/or residual cnidocytes after a sting and (2) reducing pain or symptoms. Despite the importance, the scientific community remains in disagreement on which are the best rinse solutions or treatments to provide appropriate care, which is confusing for medical professionals, lifeguards, and the general public [7,12,29,31,32,33]. Seawater, vinegar or acetic acid, freshwater, and/or baking soda are commonly used as topical rinsing solutions [7], despite the fact that they are not always the best option depending on the stinging species.

In recent years, researchers have highlighted the importance of establishing species-specific first-aid protocols to avoid extrapolating results that may lead to bad care practices due to differences in cnidocyst response between classes of the Cnidaria phylum [29,33]. Although the use of vinegar has traditionally been recommended [32,34], its unequal effectiveness between species is increasingly supported by scientific evidence [29,33,35]. While vinegar is an inhibitor of cnidocyst discharge in cubozoans [29,34], its application causes immediate discharge in scyphozoans and hydrozoans [29,33,36]. Until the present study, neither its effectiveness, nor that of other compounds commonly used as rinse solutions (e.g., ammonia, freshwater, baking soda), had been tested for *A. viridis,* leaving the first-aid guidelines for Mediterranean sea anemone stings without any scientific basis [37,38,39].

In this article, we present (1) a case study of a severe toxic reaction to *A. viridis* in a young woman working as an oceanographer, and also, for the first time, (2) an evaluation of the response of *A. viridis* cnidocysts to the presence of different topical rinse solutions, including vinegar, to establish guidelines for safe first-aid protocols for its sting.

## 2. Results

### 2.1. A Severe Toxic Reaction Following Anemonia viridis Sting

On 5 March 2021, a group of oceanographers were working in an area of approximately 1 m depth in Alfacs Bay, on the southern coast of Catalonia (Spain) (Figure 1A,B). The daily tasks of the group included the handling of bivalve mollusc individuals of the species *Pinna nobilis* (Figure 2A), whose organisms are often characterized by high and diverse levels of epibionts, including sea anemones (Figure 2B).

The group was insufficiently protected, wearing only short-sleeved T-shirts and long waterproof pants. After approximately 30 min, one of the oceanographers started to experience itching and redness on her forearms which increased progressively (Figure 3A). She washed the affected area with seawater, perceiving a little relief with this. After the group finished their daily tasks, the affected area was exposed to sunshine for 3.5 h on the way back from sampling.

Once back at the research center, severe remitting pain, itching and burning sensation led her to attend the health center. The prescribed treatment included parenteral antihistamines and anti-inflammatories, oral antihistamines, and oral and topical corticosteroids, and she returned home. The next day, 10 min after the use of the topical corticosteroid cream, she had an allergic reaction—she reported an episode of her face flushing red and feeling hot. She went back to the health center, and the doctor stopped the corticosteroid cream and diagnosed an allergic reaction to an ingredient in the cream, since the patient reported having been treated with corticosteroids previously without any reaction. The new treatment consisted of a second dose of parenteral antihistamines and anti-inflammatories, oral antihistamines, and only oral corticosteroids, with a gradually decreasing dose over one week. The young woman experienced some alleviation over the first few days but soon worsened, with an increase in pain, itch, redness, and swelling in the stung area (Figure 3B). She received a third dose of antihistamines and anti-inflammatories parenterally at the health center and oral antihistamines and high-dose oral corticosteroids for one month at home. Finally, she improved slowly, and scabs began to form on the affected area. Six months after the exposure to the toxic venom of *A. viridis* (September 2021), the area had mild hyperpigmentation.

### 2.2. First-Aid Protocols: Response of Anemonia viridis Cnidocysts to Topical Rinse Solutions

Five test solutions widely used in cnidarian first-aid protocols (seawater, vinegar, ammonia, 10% baking soda mixed in seawater, and freshwater) were evaluated (Table 1).

Seawater did not stimulate cnidocyst discharge in the tentacles of *A. viridis* (Figure 4B), so it was classified as a neutral rinse solution (Table 1). In contrast, the application of rinse solutions of vinegar and ammonia produced significant cnidocyst discharge (Figure 4C,D). The application of a solution of 10% baking soda mixed in seawater triggered a medium level of discharge (Figure 4E) compared to the immediate and massive discharge in vinegar and ammonia (Figure 4C,D). Freshwater produced a low level of cnidocyst discharge, and isolated undischarged cnidocytes were commonly observed (Figure 4F). Therefore, all the compounds except seawater were classified as activator rinse solutions, although with different degrees of discharge intensity (Table 1).

## 3. Discussion

In the Mediterranean Sea, cnidarian envenomation commonly causes local cutaneous reactions after accidental skin contact; sting episodes with systemic manifestations, including anaphylactic reactions, may also occur [15,16,17]. Although the Mediterranean sea anemone sting causes mild symptoms that do not require hospitalization, severe, isolated cases in children have triggered symptoms such as fever and loss of appetite or mobility [17]. In the present case report, the young oceanographer experienced a severe reaction to an *A. viridis* sting that worsened in the consecutive weeks. Cutaneous lesions after cnidarian envenomation can be recurrent, delayed, persistent, and occur in different body areas distant from the primary sting [15,37,40,41,42]

Marine knowledge has been reported in several occasions as scarce [43,44,45], particularly regarding knowledge related to cnidarians [46], so it should be necessary to raise awareness about the dangers present in the ocean. Unlike other cases of envenomation in bathers [15,17,21], the accident occurred during working hours, and a lack of awareness on risk prevention training was identified. In the case reported here, the patient was able to identify the species due to her experience with marine organisms, but such a background is uncommon [17]. Specific training about venomous marine biodiversity should be included in risk prevention training. Educational initiatives (e.g., Observadores del Mar [47]) and useful identification guides (Sup), as well as the use of mobile applications that report the presence or absence of jellyfish in real time (e.g., iMedJelly app [48,49]) are successful tools that provide the best scientific knowledge necessary for the identification and awareness of the presence of cnidarians in the work area. The sting reported in the present study could have been avoided by the use of physical barriers that prevent cnidocyst penetration [50]. Wearing protective clothing should have been mandatory in an area with a high presence of sea anemones such as Alfacs Bay (Figure 2A,B). Moreover, the worker should probably have avoided sun exposure on her way back from work, which is something that likely worsened the sting, since prolonged sun exposure is contraindicated in cases of cnidarian envenomation [51]. The amount of sun exposure is an essential question in the history to obtain detailed information about the cnidarian sting [52].

Updated first-aid protocols based on scientific advances are essential tools to prevent harm. In recent decades, there has been a growth in scientific reports warning of the ineffectiveness of vinegar as a rinse solution for the treatment of cnidarian stings [29,33,35,36,53], with the exception of cubozoans where its use is effective [29,33,34]. Immediate *A. viridis* cnidocyst discharge was observed after vinegar application (Figure 4C), indicating it should not be used as a rinse solution following a snakelocks sea anemone sting. This is the first time that the effectiveness of vinegar for *A. viridis* has been evaluated, yet some guidelines in reports from the medical community have recommended the use of vinegar for cleaning the affected area [37,38,39]. Herein, we have demonstrated that vinegar application is strictly contraindicated for *A. viridis* stings due to its potential as an activating solution (Table 1), producing a worsening of the affected area. This is also the case in other species of scyphozoans and hydrozoans [29,33,36,53], including the most important Mediterranean jellyfish *P. noctiluca* [29], which is responsible for the majority of jellyfish sting cases seeking assistance from lifeguards in certain Mediterranean beaches [18].

As with vinegar, the use of ammonia has also been suggested as a solution for washing the affected area following a Mediterranean sea anemone sting, without any scientific evidence [37]. After the immediate and massive cnidocyst discharge observed in this study (Figure 4D), we report the detrimental effect of its use. In the case of baking soda, some studies have shown negligible discharge in some cubozoans, scyphozoans, and hydrozoans [31,53] with the exception of increased discharge in the scyphozoan *P. noctiluca* [54]. According to our results, the application of baking soda induced cnidocyst discharge in *A. viridis*
**(**Figure 4E), although with less intensity than the other compounds; nonetheless, its use should be contraindicated.

Rinsing with freshwater is not advised in cnidarian first-aid protocols [7,14,55] because the osmotic change can promote the activation of cnidocysts [55]. In the present study, we observed a low level of discharge (Figure 4F) and the detachment of some intact cnidocytes after freshwater application. Residual cnidocytes could roll off the skin and be activated by mechanical effects [32,34]. For this reason, the use of freshwater is not recommended for *A. viridis* stings.

The use of seawater in first-aid protocols is questionable, since although it is considered a neutral and non-inhibiting solution, it could still cause discharge by mechanical effects [31,32,34]. It is considered neutral for cnidarian species, as well as for *A. viridis* as reported here (Table 1 and Figure 4B), so its use is recommended in numerous studies [29,33,35], especially when the victim is not able to recognize the species (Appendix A). Rinsing with seawater may relieve pain after stinging, as happened in the case described here, where the patient reported a little relief. Although the best option is to wash the sting site with an inhibitory solution [29], none have been identified for *A. viridis*; therefore, despite only being neutral (Table 1), we recommend seawater use until new inhibitory substances are found, as in the case of other Mediterranean cnidarians (Figure 5 and Appendix A) [29].

After safely removing tissues and/or residual cnidocytes using the rinse solution, the next step in first-aid protocols is to attenuate local pain and inflammation while trying to prevent tissue damage, inactivate toxins from the venom, and prevent venom diffusion [12]. Again, the scientific community remains in disagreement on whether to use heat (e.g., hot-water immersion, hot packs, or hot showers) or cold (e.g., ice packs or cold-water immersion) for this step [7,56]. Although the mechanism of action is still unclear, a recent systematic review recommended the use of 45 °C water immersion of the stung area for 20 min as the best treatment, which was supported by evidence of the thermolability of marine venoms, including cnidarians, and the interference of the heat in pain transmission, thus relieving pain [56]. However, the application of cold packs is reported as having a potent analgesic effect [57]. In the case of non-life-threatening species stings involving mild to moderate pain, such as *A. viridis*, the application of cold-water or cold packs is useful and easy to perform for lifeguards or even beach users. For a better understanding of the effects of hot or cold treatments, future clinical experiments are required in Mediterranean cnidarians to help clarify their implications and update the current guidelines where necessary (Appendix A).

## 4. Conclusions

Although most sea anemone stings do not pose a risk to human health, knowing the correct guidelines in the first-aid protocols helps prevent the sting from worsening. Our research reveals the ineffectiveness—and indeed contraindication—of vinegar as a rinse solution, as well as that of ammonia, baking soda, and freshwater, for *A. viridis* sting. In addition, we strongly recommend the use of physical barriers such as stinger suits, wet-suits, or dry-suits to prevent the injection of cnidocysts for users of coastal environments, including sea workers. Avoidance of sun exposure is advised in severe stings.

The appropriate first-aid protocol after *A. viridis* sting includes the following (Figure 5): (1) Remove pieces of tentacles or other tissues with a rigid element such as a credit card or tweezers. Never rub. (2) Rinse the affected area with seawater (never vinegar, ammonia, baking soda, or freshwater). (3) Apply ice packs for 15 min, in intervals of 5 min (3 min of application and 2 min of rest), covered by a towel or cloth. (4) If pain persists, go to the nearest health center.

Efforts to improve knowledge related to cnidarians are essential and urgent in those who provide first-aid in order to revert dangerous trends that threaten users exposed to venomous organisms. First, an agreed framework is urgently needed to adopt and implement first-aid protocols for cnidarian species, as well as further development of discharge-inhibiting substances that can act as an effective rinsing solution to safely remove tissues and/or residual cnidocytes after a sting. More efforts should be made to better train lifeguards, health professionals and users of coastal environments and provide them with specific and updated materials (Appendix A).

## 5. Materials and Methods

### 5.1. Interview with the Patient

The questionnaire included the following information modified from Burnett (2001) [52]:

Details of the sting:Date, time, locationCnidarian species that caused the stingPatient activity at the time of the stingWater conditions and amount of sun exposureClothing or sun cream protectionPart of body stung

Symptoms:Local symptoms: e.g., pain, redness, papules, swellingSystemic symptoms: e.g., fever, muscle spasms, nausea

Past medical history:Past cnidarian stingSignificant medical diseasesDrugs taken at the time of the stingAllergy history: e.g., asthma, hay feverAny current medications

First-aid protocols:Did you notice any remains of cnidarian tissue attached after the sting? How did you remove them?First-aid protocol for rinsing the sting area: e.g., use of seawater, vinegarFirst-aid protocol for relieving pain: e.g., hot or cold applicationDid you know the proper first-aid protocol for cnidarian stings? (“cnidarian” includes animals such as jellyfish, sea anemone, and coral)Had you previously received any information on how to act in case of a cnidarian sting during occupational risk training?Did you need medical assistance? e.g., time of arrival at health center, number of visits, treatment, outcome

Other information:Any other information that you would like to provide?

The patient received and signed a consent form before the interview.

### 5.2. First-Aid Protocol Experiments

#### 5.2.1. Sea Anemone Collection

*Anemonia viridis* specimens were collected, at a depth of 1 to 5 m, during scuba dive sampling in September 2021 along the Calp coast (Alicante, Spain) (Figure 1C and Figure 2C). To ensure the integrity of the tentacles, whole individuals were collected using spatulas and transferred to receptacles full of seawater, avoiding air bubbles inside in order to maintain the organisms in the best conditions. Then, the specimens were transported to the Institute of Environmental Research and Marine Science (IMEDMAR-UCV) laboratories for maintenance, where they were kept in aquarium tanks without feeding until their use (less than 48 h).

#### 5.2.2. Tentacle Solution Assay

The evaluation of cnidocyst discharge was tested in five rinse solutions: seawater, vinegar (Vivó, 6% acetic acid), ammonia (Hacendado), 10% baking soda (Hacendado) in seawater, and freshwater (Table 1) [34].

Tentacle pieces of approximately 3 cm long from 8 individuals of *A. viridis* were transferred to slides (76 × 26 cm). The samples were observed under the light microscope to ensure their integrity (Figure 4A). Then, 15 µL of each solution were applied to the tentacle to determine the effect on discharge during a period of 30 s. As defined by Ballesteros et al. (2021) [29], the study area was the rim of the tentacle.

The cnidocyst response was classified qualitatively into four categories [33]:0: no discharge observed;+: low discharge of cnidocysts;++: medium discharge of cnidocysts;+++: high discharge of cnidocysts.

The effect of the rinse solution was classified into one of the following two categories:Activator effect solution: cnidocysts were activated after the application of the solution;Neutral effect solution: cnidocysts were not activated after the application of the solution.

## Figures and Tables

**Figure 1 toxins-14-00027-f001:**
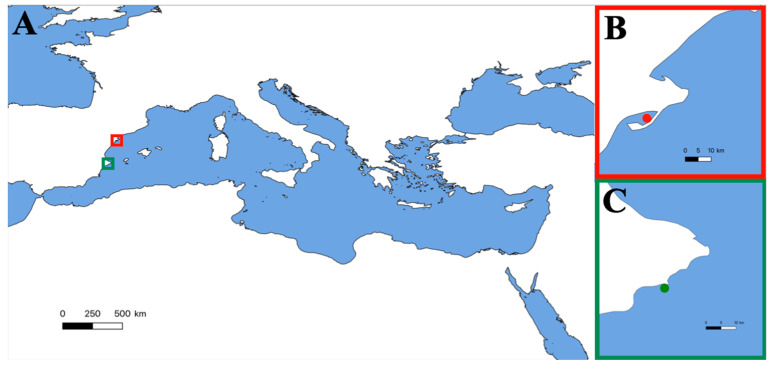
(**A**) Mediterranean Sea. (**B**) Alfacs Bay (north-western Mediterranean). Location of the interaction between the patient and the sea anemone (red dot). (**C**) Sample collection of *Anemonia viridis* for first-aid protocol experiments (green dot).

**Figure 2 toxins-14-00027-f002:**
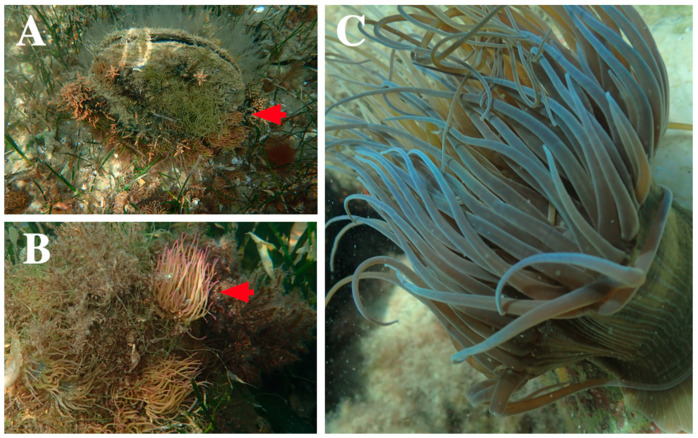
(**A**,**B**) Individuals of the species *Pinna nobilis* covered with high densities of sea anemone (red arrows) in Alfacs Bay (Spain). (**C**) *Anemonia viridis* in low bathymetric area sampling in Calp (Spain).

**Figure 3 toxins-14-00027-f003:**
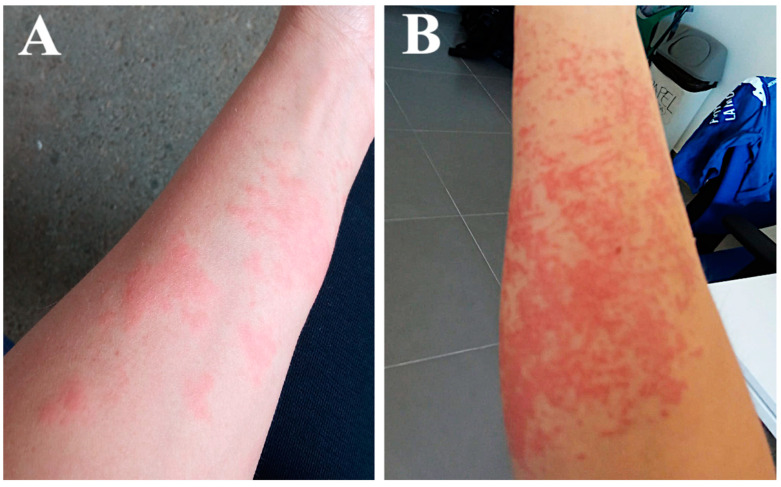
(**A**) *Anemonia viridis* sting sustained by the patient during field work. (**B**) Worsening of the affected area.

**Figure 4 toxins-14-00027-f004:**
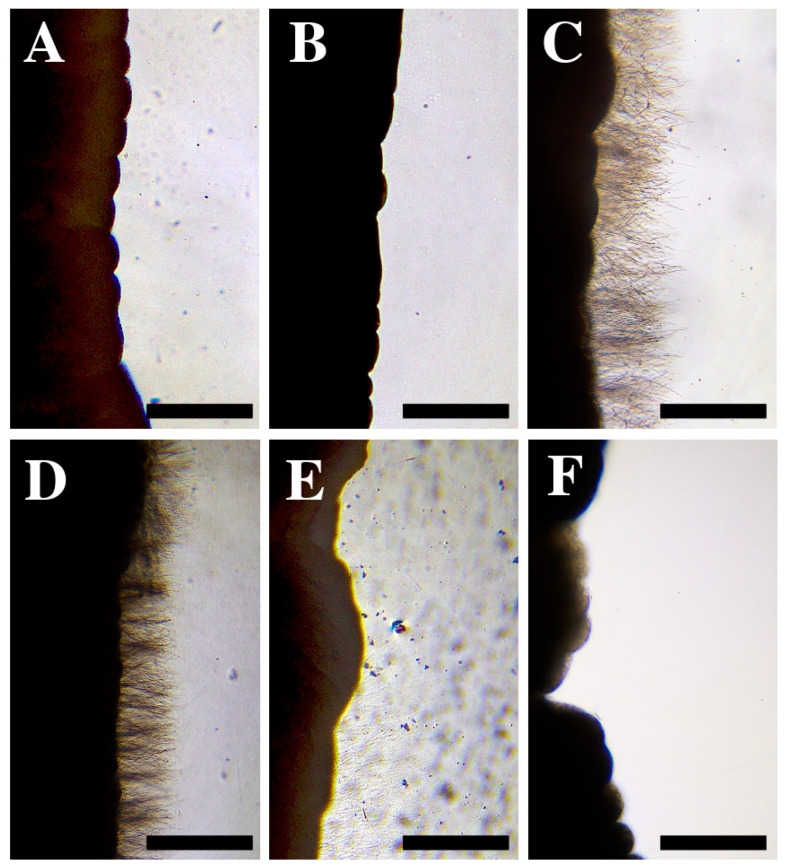
(**A**) Tentacle pieces of *Anemonia viridis* without any treatment. (**B**–**F**) Cnidocyst responses after the application of: (**B**) seawater, (**C**) vinegar, (**D**) ammonia, (**E**) 10% baking soda mixed in seawater, and (**F**) freshwater. Note the different response between a neutral solution (**B**) and an activator solution (**C**–**F**). Scale bars: 0.5 mm.

**Figure 5 toxins-14-00027-f005:**
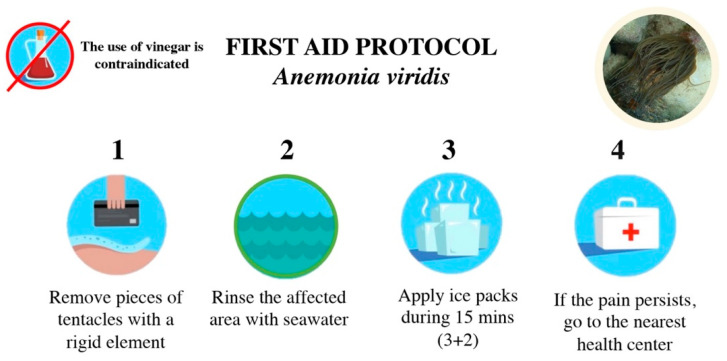
First-aid protocol for stings by the sea anemone *Anemonia viridis*.

**Table 1 toxins-14-00027-t001:** *Anemonia viridis* cnidocyst response after the application of different rinse solutions.

*Anemonia viridis*			
Rinse Solutions	*n*	Discharge ^1^	Effect ^2^
Seawater	20	0	Neutral
Vinegar	20	+++	Activator
Ammonia	20	+++	Activator
10% Baking soda mixed in seawater	20	++	Activator
Freshwater	20	+	Activator

Method: Tentacle Solution Assay. ^1^ Cnidocyst discharge categories: 0 = no discharge of cnidocysts; + = low discharge of cnidocysts; ++ = medium discharge of cnidocysts; +++ = high discharge of cnidocysts. ^2^ Rinse solution categories: neutral solution = cnidocysts are not activated after the application of the solution; activator solution = cnidocysts are activated after the application of the solution; *n* indicates the number of replicates.

## Data Availability

Data sharing not applicable. No new data were created or analyzed in this study.

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
