# Peer review of "Trial Assay for Safe First-Aid Protocol for the Stinging Sea Anemone Anemonia viridis (Cnidaria: Anthozoa) and a Severe Toxic Reaction"

_toxins, 2022, doi:10.3390/toxins14010027_

Round 1

Reviewer 1 Report

Authors reported a case of envenoming by Anemonia viridis, an anemone that is abundant and widely-distributed in temperate sea which can form dense congregations of individuals. Its sting is potentially severe, but only a few detailed cases have been reported. The description of the toxic reaction, risk factors and treatment in the case is valuable to the literature. The authors further conducted a research to test the Anemonia viridis cnidocyst response after the application of different rinse solutions. They applied the finding from the study to develop a species-specific first-aid protocol, which will benefit the public and community. The illustration and developed first aid brochure are interesting and useful. This paper is well written and can be considered for publication in Toxins. There is probably one query which the authors may clarify or elaborate in the manuscript: As species-specific first aid is important (hence, the value of this study), how can the public or paramedic identify the stinging species confidently in the field? Would there be any possibility that a patient is stung by other species or even multiple species at one time? Also, this protocol/first aid is applicable to which area - presumably only in places where the species is distributed? Consequently, it should be mentioned as if it is to be used for envenoming by another species, will it possibly cause more harm instead? 

Author Response

Response to Reviewer 1 Comments

  1. How can the public or paramedic identify the stinging species confidently in the field?

Identifying the species in the field is difficult if the victim is not able to recognize it or has not seen the specimen. For example, in Catalonia, we work on the training of lifeguards and provide them with identification cards so that they or the victim can recognize the species when they go to the first aid post and from there, apply the appropriate specific protocol. 

  1. Would there be any possibility that a patient is stung by other species or even multiple species at one time?

It can happen, but it usually doesn't happen.

  1. Also, this protocol/first aid is applicable to which area - presumably only in places where the species is distributed?

Yes. To avoid extrapolation of results from other species, in our opinion, it is better to refer only to the species. The problem of using vinegar traditionally is due to extrapolating the data from first aid protocols from other areas, such as Australia, where species of box jellyfish dominate and vinegar is effective. For example, when we have investigated Mediterranean species, such as Pelagia noctiluca, we have observed that vinegar produce nematocyst discharge. Also for other scyphozoans such as Rhizostoma luteum (unpublished data). Therefore, in our opinion, it is better to indicate specific-species and not areas.

  1. Consequently, it should be mentioned as if it is to be used for envenoming by another species, will it possibly cause more harm instead?

Seawater is considered a neutral compound, although it was used by other species it would not be a problem. The problem is activated solutions, such as vinegar, which if used in species where discharge can worsen the sting.

Reviewer 2 Report

This manuscript documents a specific case of a sea anemone (Anemonia viridis) envenomation along with a first-aid treatment assessment. This manuscript importantly highlights the differences between appropriate treatments in Cnidaria. Specifically, no two cnidarians are the same and so species specific treatment forms invaluable information for the first-aid responders and clinicians.  

My recommendation is publication after minor corrections. 

Specific points are as follows 

  1. Cnidarians are “venomous” not “poisonous”, please amend. 
  2. To my knowledge no Actiniaria (True sea-anemones) contain ptychocysts and it is only in cerianthids. Please amend statement. Additionally, subclass would be discussed prior to an Order 
  3. Terminology throughout the paper needs to be corrected. The cnidocyst is the gland in which the organelle, the cnidae (nematocyst, spirocyst and ptychocyst) is produced. It is the cnidae which fires and injects the venom. Not the cnidocyst. It is the cnidae which are classified as nematocysts, spirocysts and ptychocysts. Please correct. 
  4. The term benthic is used incorrectly – benthic means ocean bottom – Anemonia viridis is an example of intertidal or shallow subtidal sea anemone. Not all sea anemones are benthic i.e. found on the sea floor. Please amend. 
  5. Cnidome is spelt without an e – cnidom and is the full complement of cnidae not cnidocysts 
  6. In the heading 2.2. because the entire heading is italicised the correct format is to unitalicise the species name e.g. The heading would be formatted as such First-aid protocols: Response of Anemonia viridis cnidocysts to topical rinse solutions. 
  7. The phrase “Low levels of ocean literacy” does not make sense nor is it grammatically correct. Please clarify. I believe, the authors are trying to convey societies awareness of the dangers present in the ocean. 
  8. At what depth were specimens collected and was it under a permit. If so, state the permit for collection. Please state the specialist who taxonomically identified the animals. 
  9. Recipients should be receptacles. A recipient is a person not a container. 
  10. Reference 56 requires a capital H for heated. 
  11. Can the authors please clarify if the cold packs should have a covering on application or if cold packs are placed directly onto the envenomation site. That is, would placing the ice pack directly onto the site without a covering result in condensation being formed and potentially causing residual nematocysts to fire? 
  12. Was ethics approval obtained for the interview with the envenomation victim. 
  13.  

Author Response

Response to Reviewer 2 Comments

  1. Cnidarians are “venomous” not “poisonous”, please amend. 

Done.

“Cnidarians are recognized as venomous marine animals.”

  1. To my knowledge no Actiniaria (True sea-anemones) contain ptychocysts and it is only in cerianthids. Please amend statement. Additionally, subclass would be discussed prior to an Order 

Done

“Spirocysts are used to immobilize prey in most anthozoans, and ptychocysts are used by the the subclass Ceriantharia for the creation of protective tubes to live in [4,6].”

  1. Terminology throughout the paper needs to be corrected. The cnidocyst is the gland in which the organelle, the cnidae (nematocyst, spirocyst and ptychocyst) is produced. It is the cnidae which fires and injects the venom. Not the cnidocyst. It is the cnidae which are classified as nematocysts, spirocysts and ptychocysts. Please correct. 

Thank you very much for the comment.

According to the bibliography (referenced below), the stinging cell is the cnidocyte. From the Golgi apparatus, a subcellular-enclosed capsule, known as a cnidocyst (or cnidae), is formed with an everted tubule along with a mixture of toxins. Cnidocysts are classified into three categories: nematocyst, spirocyst and ptychocyst.

Cnidae are secreted by the Golgi apparatus of all cnidarians and only cnidarians. Of the three categories of cnidae (also called cnidocysts), nematocysts occur in all cnidarians, and are the means by which cnidarians defend themselves and obtain prey; spirocysts and ptychocysts are restricted to a minority of major taxa (Faustin et al., 2009).

Fautin, D. G. (2009). Structural diversity, systematics, and evolution of cnidae. Toxicon, 54(8), 1054-1064.

Östman, C. (2000). A guideline to nematocyst nomenclature and classification, and some notes on the systematic vaue of nematocysts. Scientia Marina, 64(S1), 31-46.

Strömberg, S. M., & Östman, C. (2017). The cnidome and internal morphology of Lophelia pertusa (Linnaeus, 1758)(Cnidaria, Anthozoa). Acta Zoologica, 98(2), 191-213.

The term cnidocyst discharge is used for many papers (referenced below). In this case, we use cnidocyst discharge encompassing the nematocyst and spirocyst categories of Anemonia viridis.

Dahl Hermansen, T., Arvedlund, M., & Fiedler, G. C. (2005). Calcium antagonists inhibit the discharge of cnidae in response to electrical stimulation in the giant tropical sea anemone Heteractis crispa Ehrenberger (Anthozoa). Marine and Freshwater Behaviour and Physiology, 38(4), 269-274.

Turk, T., & Kem, W. R. (2009). The phylum Cnidaria and investigations of its toxins and venoms until 1990. Toxicon, 54(8), 1031-1037.

Helmholz, H., Ruhnau, C., Schütt, C., & Prange, A. (2007). Comparative study on the cell toxicity and enzymatic activity of two northern scyphozoan species Cyanea capillata (L.) and Cyanea lamarckii (Péron & Léslieur). Toxicon, 50(1), 53-64.

  1. The term benthic is used incorrectly – benthic means ocean bottom – Anemonia viridis is an example of intertidal or shallow subtidal sea anemone. Not all sea anemones are benthic i.e. found on the sea floor. Please amend. 

Done.

“However, other cnidarians, such as sea anemones, are also responsible for severe skin reactions in the Mediterranean basin.”

  1. Cnidome is spelt without an e – cnidom and is the full complement of cnidae not cnidocysts 

Thank you very much for the comment.

The term cnidome (with e) is widely used in many papers, and cnidocyst is synonoum of cnidae. In our opinion, referring to the cnidome as a set of cnidocytes also is correct, but we can change it to cnidocysts (they are categories of the taxonomic classification).

“The cnidome, which includes the total complement of cnidocysts within a cnidarian specimen [4,5], comprises four types in adult individuals collected in the North Adriatic Sea; three nematocysts (holotrichous isorhiza, p-mastigophore and a possible microbasic b-mastigophore) and one spirocyst [26].”

Di Camillo, C., Bo, M., Puce, S., Tazioli, S., & Bavestrello, G. (2006). The cnidome of Carybdea marsupialis (Cnidaria: Cubomedusae) from the Adriatic Sea. Journal of the Marine Biological Association of the United kingdom, 86(4), 705-709.

Strömberg, S. M., Östman, C., & Larsson, A. I. (2019). The cnidome and ultrastructural morphology of late planulae in Lophelia pertusa (Linnaeus, 1758)—With implications for settling competency. Acta Zoologica, 100(4), 431-450.

Ballesteros, A., Östman, C., Santín, A., Marambio, M., Narda, M., & Gili, J. M. (2021). Cnidome and Morphological Features of Pelagia noctiluca (Cnidaria: Scyphozoa) Throughout the Different Life Cycle Stages. Frontiers in Marine Science, 1059.

Avian, M., & Malej, A. (2015). Aurelia polyps and medusae (Scyphozoa; Semaeostomeae; Ulmaridae) in the Northern Adriatic: their cnidome and ecology. In PERSEUS International Workshop “Coming to grips with the jellyfish phenomenon in the Southern European and other Seas: research to the rescue of coastal managers.” (pp. 17-17). PERSEUS.

  1. In the heading 2.2. because the entire heading is italicised the correct format is to unitalicise the species name e.g. The heading would be formatted as such First-aid protocols: Response of Anemonia viridis cnidocysts to topical rinse solutions. 

Thank you very much for the comment.

In the latest version in this document for us the titles are shown like this:

2.2. First-aid protocols: Response of Anemonia viridis cnidocysts to topical rinse solutions

I will ask the editor to take it into account in the final version.

  1. The phrase “Low levels of ocean literacy” does not make sense nor is it grammatically correct. Please clarify. I believe, the authors are trying to convey societies awareness of the dangers present in the ocean. 

Done.  We have removed the term ocean literacy.

“Marine knowledge has been reported in several occasions as scarce [43–45], particularly regarding knowledge related to cnidarians [46], so it should be necessary to raise awareness about the dangers present in the ocean.”

“Efforts to improve knowledge related to cnidarians are essential and urgent in those who provide first-aid in order to revert dangerous trends that threaten users exposed to venomous organisms.”

  1. At what depth were specimens collected and was it under a permit. If so, state the permit for collection. Please state the specialist who taxonomically identified the animals. 

Done.

The depth at which the individuals were collected was 1 to 5 meters deep.

Anemonia viridis specimens were collected, at a depth of 1 to 5 meters, during scuba dive sampling in September 2021 along the Calp coast (Alicante, Spain) (Figure 1C, 2C).”

No permission was needed.

The taxologist who identified the specimens was Dr. Josep-Maria Gili, co-author of this work (author contribution)

“J.-M.G taxonomically identified the specimens.”

  1. Recipients should be receptacles. A recipient is a person not a container. 

Done.

“To ensure the integrity of the tentacles, whole individuals were collected using spatulas and transferred to receptacles full of seawater, avoiding air bubbles inside in order to maintain the organisms in the best conditions.”

  1. Reference 56 requires a capital H for heated. 

Done.

Wilcox, C.; Yanagihara, A. Heated debates: hot-water immersion or ice packs as first aid for cnidarian envenomations? Toxins 2016, 8, 97, doi:10.3390/toxins8040097.

  1. Can the authors please clarify if the cold packs should have a covering on application or if cold packs are placed directly onto the envenomation site. That is, would placing the ice pack directly onto the site without a covering resul in condensation being formed and potentially causing residual nematocysts to fire? 

Yes. Done.

“(3) Apply ice packs for 15 min, in intervals of 5 min (3 min of application and 2 min of rest), covered by a towel or cloth.”

When cold packs are applied, residual cnidocytes have already been removed by the rinse solution, therefore it should not cause any discharge.

  1. Was ethics approval obtained for the interview with the envenomation victim. 

Yes.

A protocol to ensure confidentiality was followed. A consent form with all the terms and conditions as well as rights to revoke information provided was provided to the patient and signed. This document has been sent to the editor.